# Whey Protein Hydrolysate Improved the Structure and Function of Myofibrillar Protein in Ground Pork during Repeated Freeze–Thaw Cycles

**DOI:** 10.3390/foods12163135

**Published:** 2023-08-21

**Authors:** Pengjuan Yu, Jiayan Yan, Lingru Kong, Juan Yu, Xinxin Zhao, Xinyan Peng

**Affiliations:** 1College of Life Sciences, Yantai University, Yantai 264005, China; yupengjuan99@163.com (P.Y.); yanjiayan2023@163.com (J.Y.); konglr2226@163.com (L.K.); yujuan14615@163.com (J.Y.); 2College of Food Science and Engineering, Yangzhou University, Yangzhou 225127, China; xxzhao@yzu.edu.cn

**Keywords:** whey protein hydrolysate, repeated freeze–thaw, ground pork, myofibrillar protein

## Abstract

Whey protein hydrolysate (WPH) has made a breakthrough in inhibiting oxidative deterioration and improving the quality of meat products during storage. Based on our previous study of extracting the most antioxidant active fraction I (FI, the molecular weight < 1 kDa) from whey protein hydrolysates of different molecular weights, the present study continued to delve into the effects of WPH with fraction I on the structure and function of myofibrillar proteins (MP) in ground pork during the freeze–thaw (F-T) cycles. With the number of F-T cycles raised, the total sulfhydryl content, the relative contents of α-helix, Ca^2+^-ATPase activity, K^+^-ATPase activity, solubility, emulsion activity index (EAI), and emulsion stability index (ESI) of MP gradually decreased. Conversely, the carbonyl content and the relative content of random curl showed an increasing trend. In particular, the damage to the structure and the function of MP became more pronounced after three F-T cycles. But, during F-T cycles, FI stabilized the structure of MP. Compared to the control group, the 10% FI group showed a remarkable improvement (*p* < 0.05) in the total sulfhydryl content, Ca^2+^-ATPase activity, K^+^-ATPase activity, solubility, EAI and ESI after multiple F-T cycles, suggesting that 10% FI could effectively inhibit protein oxidation and had the influence of preserving MP function properties. In conclusion, WPH with fraction I can be used as a potential natural antioxidant peptide for maintaining the quality of frozen processed meat products.

## 1. Introduction

A popular method for long-term storage of meat products is freezing preservation, which maintains the quality of meat products and extends their sell-by date by limiting microbial growth [1]. However, due to the faultiness of the cold-chain system and the inconsistent practices among sellers and consumers in daily life, multiple freeze–thaw (F-T) cycles inevitably occur during the storage, production, transit, and sale of meat products [2], which could contribute to the production of poor color, texture deterioration, flavor loss, and water loss of the product, with serious economic losses [3,4,5].

Myofibrillar proteins (MP) in muscle tissue are closely linked to the functional properties of the protein in meat products (such as emulsification, water retention, and gelation properties), also playing a crucial role in enhancing the texture and moisture retention of meat products [6]. Consequently, the denaturation and degradation of muscle proteins are the main causes contributing to the adverse alterations in the physical and chemical properties of meat products induced by the F-T cycles [7]. However, due to changes in the size and orientation of ice crystals during the F-T cycles, the myofibrillar tissue is destroyed and the connective tissues are loosened, thereby diminishing the protein stability in muscle [8]. Furthermore, the squeezing action exerted by ice crystals increases protein oxidation, further promoting the unfolding of protein structures that leads to the generation of free radicals, the exposure of hydrophobic structural domains, the production of amino acid derivatives, and the breakdown in protein structures [9]. It has been demonstrated that extensive extracellular ice crystal formation throughout repeated F-T processes alters the physical structure of myofibrils and internal myogenic fibers, causing MP denaturation and aggregation, which leads to the collapse and disintegration of protein structures [10]. Therefore, it is crucial to stabilize the structure and improve the functional properties of MP for producing high-quality ground pork products and extending their shelf life during the F-T cycles.

Currently, to minimize protein oxidation, denaturation, and aggregation, researchers are working on the use of antioxidant additives to prevent or improve quality-related changes in frozen meat products during the F-T cycles [1]. In particular, the food-derived bioactive peptides, such as whey protein hydrolysate (WPH) [11], silver carp scales antifreeze peptides [12], chicken skin collagen hydrolysate [13], and so forth, were extensively used as the antioxidant additives due to their high activities and low toxicity. Among them, WPH is generated by enzymatic hydrolysis of natural whey protein (NWP), which possesses a high content of peptides and amino acids. Due to its low molecular weight, WPH is frequently employed as a nutritious and bioactive ingredient in food and nutraceutical fields and exhibits good antioxidant, anti-fatigue, and antihypertensive properties [14]. It has been shown that WPH has made a breakthrough in enhancing the mass of meat products. Vavrusova et al. [15] issued the potential of WPH as food antioxidants to inhibit oxidative deterioration during the storage of meat products. Peng et al. [16] found that WPH (<3 kDa) exhibited superior antioxidant activity, thereby effectively inhibiting oxidation and improving the gel quality of ground meat.

Based on our previous research, fraction I (FI < 1 kDa) with excellent hygroscopicity, solubility, and stable spatial conformation was found in WPH, and exhibited the most significant oxidation resistance. The addition of FI to pork patties was found to significantly improve the cooking loss and effectively reduced the destruction of the microstructure of frozen–thawed pork patties [17]. So, it is reasonable to assume that FI has an inhibitory effect on fat and protein oxidation in the meat system. However, the impact of FI on the attributes of structure and function of MP after interacting with the protein is not clear, necessitating further exploration. Therefore, the aim of this study was to investigate the effect of FI on MP oxidation and structure in ground pork after multiple F-T cycles by determining the carbonyl content, total sulfhydryl content, Ca^2+^-ATPase activity, K^+^-ATPase activity, and secondary structure content, and then to explore the protective mechanism of FI on the structure and function of MP by examining the solubility and emulsification properties. This study can be expected to reveal the effect of WPH with fraction I on the structure and function of MP in ground pork during F-T cycles, and further facilitate the development of WPH with fraction I as potential natural antioxidant peptides to prevent quality-related deterioration, extend shelf life, and produce high-quality ground meat products.

## 2. Materials and Methods

### 2.1. Chemicals and Materials

NWP (95%) originated from Davisco Foods International, Inc. (Le Sueur, MN, USA). Fresh pork longissimus dorsi and back fat were purchased from a supermarket located in Yantai (Yantai, China). The chemical reagents used in the experiment were partially purchased from Sigma Chemical Inc. (St. Louis, MO, USA) and included: alkaline enzyme (6 × 10^4^ U/g), phosphate buffer, butylated hydroxy anisole (BHA), 5,5′-Dithiobis(2-nitrobenzoic acid) (DTNB), and ethylene diamine tetra acetic acid (EDTA). Other chemical reagents were purchased from Sinopharm Chemical Reagent Co., Ltd. (Shanghai, China).

### 2.2. Preparation of FI

The method described by Peng et al. [18] was used to make WPH, with a few minor changes. Initially, NWP was dissolved into a solution (10 g/100 mL) and then preheated in a water bath for five minutes at 95 °C. Subsequently, NaOH (1 mol/L) was added to maintain the pH of the protein solution at 8.5. The pH was adjusted to 7.0 after NWP was hydrolyzed by Alcalase with an enzyme-to-protein (E/S) ratio of 2:100 at 65 °C for eight hours. Subsequently, the solution was heated in a boiling water bath for ten minutes to inactivate the enzyme, and lyophilized it to obtain the hydrolysis products (WPH). The antioxidant activity of WPH at different hydrolysis times was determined by measuring its ability to scavenge 1-diphenyl-2-picrylhydrazyl (DPPH) radicals, which exhibited a value of 82.62% after four hours of hydrolysis time. Furthermore, after ultrafiltration, the ferric-reducing antioxidant power (FRAP) assay revealed that FI (<1 kDa) possessed the highest reduction ability among hydrolysates with different molecular weights, exhibiting a value of 1274.2 μmol/L FRAP capacity. Therefore, the FI fraction was selected for subsequent experiments.

### 2.3. Preparation of Ground Pork

The pork longissimus dorsi was trimmed to remove visible connective and fatty tissues. Next, the pork longissimus dorsi and back fat were grounded in a meat grinder at a ratio of 3:7. Afterwards, the pork mince was randomly and equally divided into six groups with equal amounts of 500 g per group, and treated as follows: the control group was treated without any additives, while the other five groups received 10% NWP, 5% FI, 10% FI, 15% FI, or 0.02% BHA (positive control). Furthermore, each treatment group was supplemented with 1.5% (*w/v*) NaCl solution. After thorough mixing, each group (75 g) was made up and individually packed in polyethylene bags. Throughout the manufacturing of ground pork, the temperature was kept at approximately 4 °C. All ground pork samples were frozen for five days at −18 °C, then thawed for 12 h at 4 °C until the central temperature reached 0~2 °C, which represented one F-T cycle. In accordance with the aforementioned process, 3, 5, and 7 F-T cycles were carried out.

### 2.4. Extraction of MP

The MP was extracted according to the method of Xue et al. [19] with minor modifications. The ground pork was homogenized with 4 volumes (*w/v*) of the isolation buffer (0.1 mol/L NaCl, 1 mmol/L EDTA, 2 mmol/L MgCl_2_, and 10 mmol/L K_2_HPO_4_; pH 7.0), filtered through four layers of coarse cotton cloth, and then centrifuged at 3500× *g* for 15 min at 4 °C to discard the supernatant. The pellet was extracted 2 more times with the same isolation buffer, and then washed twice with 0.1 mol/L KCl. Subsequently, the MP concentration was determined using the Biuret method.

### 2.5. Carbonyl Content Determination of MP

The MP’s carbonyl content was determined according to the method of Chen et al. [20] with a few minor adjustments. In brief, 0.5 mL of MP solution (40 mg/mL) was mixed with the same volume of 0.2% 2,4-dinitrophenyl hydrazine (DNPH) in 2 mol/L HCl for 1 h at room temperature. Subsequently, 2 mL of 10% trichloroacetic acid (TCA) was added, and the supernatant was removed by centrifuging at 8500× *g* for 10 min at 4 °C. The obtained pellets were washed twice with 3 mL of ethyl/ethyl acetate (1:1, *v/v*), and then dissolved thoroughly in 2 mL of 6 mol/L guanidine hydrochloride for 15 min at 37 °C. The absorbance values were measured at 370 nm following a 10-min period of centrifugation at 8500× *g* for 3min.

### 2.6. Total Sulfhydryl (SH) Content Determination of MP

The total SH content of MP was measured based on the method used by Wang et al. [21] with minor adjustments. Briefly, 9 mL of Tris-HCl buffer (8 mol/L urea, 2% sodium dodecyl sulfate, and 10 mmol/L EDTA; pH 8.0) was added to 1 mL of MP solution. A total of 4 mL of the above-mixed solution was then added to Tris-HCl buffer (pH 8.0) with 0.1% DTNB, followed by incubation at 40 °C for 10 min. The absorbance values were measured at 412 nm.

### 2.7. Ca^2+^-ATPase and K^+^-ATPase Activities of MP

The ATPase activities of MP were determined following the method described by Xia et al. [22] with a few minor modifications. Firstly, 0.2 mL of MP samples (3.0 mg/mL) were mixed with 2.0 mL reaction solution including 15 mmol/L CaCl_2_, 150 mmol/L KCl, 7.6 mmol/L ATP, and 180 mmol/L Tris-HCl at pH 7.4 for the detection of Ca^2+^-ATPase activity, as well as 300 mmol/L KCl, 5 mmol/L EDTA, 7.6 mmol/L ATP, and 180 mmol/L Tris-HCl at pH 7.4 for the measurement of K^+^-ATPase activity. After incubation at 25 °C for 10 min, the reaction was stopped by adding 10% TCA (1 mL). The supernatant was collected by centrifuging at 2500× *g* for 5 min, and mixed with 3 mL of ammonium molybdate (6.6 g/L) and 0.75 mol/L of H_2_SO_4_. Then, 0.5 mL of freshly prepared FeSO_4_ solution (10%) was added to the above mixture and reacted for 2 min. To evaluate the Ca^2+^-ATPase and K^+^-ATPase activities, the absorbance values of the liberated inorganic phosphate were measured at 700 nm. The results were converted to μmol phosphate/mg MP.

### 2.8. Secondary Structure of MP

According to the method of Qian et al. [23] with minor modifications, the MP’s secondary structure was measured using the Fourier transform infrared spectrometer (Thermo, Nicolet iS20, Waltham, MA, USA). MP solution was initially diluted to 2.0 mg/mL, and the spectra of the samples in the 4000–400 cm^−1^ region with a frequency of 32 and a resolution of 4 cm^−1^ were recorded using the OMNIC 8.2 software, which comes with the infrared spectrometer. The detected data were then subjected to Fourier deconvolution.

### 2.9. The Protein Solubility of MP

The protein solubility of MP was assessed following the method of Hu et al. [24]. A total of 5 mL of MP solutions was centrifuged at 10,000× *g* for ten min, and the protein concentration of the supernatant before and after centrifugation was determined by the Bradford method using a spectrophotometer (UV-2102C; Unico, Wixom, WI, USA). The ratio of the protein content in the supernatant after centrifugation (C_after_) to the initial content in the sample (C_before_) was defined as the protein solubility as follows:$$\mathrm{Protein}\mathrm{solubility}\left(\%\right)=\frac{C_{\mathrm{after}}}{C_{\mathrm{before}}}\times 100$$

### 2.10. Emulsifying Properties

The emulsifying performance of MP was assessed as described by Du et al. [25]. To prepare the emulsion, 2.0 mL of pure corn oil and 8.0 mL of a 10 mg/mL MP protein solution were mixed in a beaker, and homogenized at 6000 rpm for 1 min. The 50 μL sample of the emulsion was collected from 0.5 cm above the beaker’s bottom and dispersed in 5 mL of the solution with 0.1% sodium dodecyl sulfate (SDS) solution. The 0.1% SDS solution was used as a blank control, and the absorbance values of the emulsion were measured at 500 nm using a spectrophotometer (UV-2102C; Unico, USA). After allowing the emulsion to stand at 23 °C for ten min, the absorbance values were measured again at 500 nm. The emulsion activity index (EAI) and emulsion stability index (ESI) were calculated as follows:$$\mathrm{EAI}\;(m^{2}/g)=\frac{2\times 2.303}{C\times (1-\varphi )\times 10^{4}}\times A_{0}\times N$$
where A_0_ was the absorbance at 500 nm; C represented the protein concentration (g/mL); φ was the oil volume fraction (*v/v*) of the emulsion (0.20); and N mean dilution factor;
$$\mathrm{ESI}\;(\%)=\frac{A_{10}}{A_{0}}\times 100$$
where A_0_ and A_10_ were the absorbance at 500 nm at time zero and 10 min, respectively.

### 2.11. Statistical Analysis

Three separate batches of ground pork were prepared (repeated), and each batch of ground pork was determined in triplicate. Data were analyzed statistically using the Statistix 8.1 software (Statistix 8.1, Analytical Software, St Paul, MN, USA), and presented as the mean ± standard deviations (SD). SPSS Statistics 16.0 was required to test the data for normality and variance homogeneity. The analysis of variance (ANOVA) with Tukey’s multiple comparisons was used to measure the significance of the treatment effects (*p* < 0.05).

## 3. Results and Discussion

### 3.1. Carbonyl Content of MP

The variations in the carbonyl content of meat protein are an early marker of oxidative modification of proteins by a variety of amino acids. Certain amino acids, such as lysine, arginine, and proline, are easily converted to carbonyl groups during oxidation [26]. Therefore, evaluating the degree of oxidation in MP is often accomplished by measuring the change in carbonyl content. According to Figure 1, with the increasing F-T cycles, the carbonyl contents in each group showed an increasing trend and displayed a significant enhancement, especially after three cycles. It is evident that meat protein easily forms carbonyl compounds during the oxidation process, and the freezing process is crucial for the formation of protein carbonyl. This can be attributed to the disruption in the microstructure and integrity of muscle cells by the water recrystallization occurring during repeated F-T cycles. These physical changes in the muscle cells promote the release of pro-oxidants, thus exacerbating the protein oxidation and accelerating the formation of carbonyl groups [27]. Ice crystal growth also results in an increase in the concentration of solutes within the cell, which can further concentrate pro-oxidants and accelerate oxidative denaturation of proteins, thereby promoting the production of carbonyl groups [28]. What is more, the primary and secondary lipid peroxides (malondialdehyde) can probably interact with MP to generate carbonyl groups [29]. After seven F-T cycles, the control group exhibited the highest carbonylation level at 1.58 nmol/mg, which was substantially different from FI and BHA addition groups (*p* < 0.05). At the five and seven F-T cycles, the carbonyl contents of the 10% FI and 0.02% BHA groups were considerably lower than the other four groups (*p* < 0.05). These findings indicate different variations in protein oxidation among each group. This may be due to the high antioxidant activity of FI and BHA, which effectively inhibit protein oxidation by scavenging reactive oxygen radicals, inhibiting the production of lipid peroxides, as well as preventing the formation of carbonyl groups in MP. It is evident that the addition of FI to ground pork may impede the growth of ice crystals by interacting with MP, thereby mitigating freezing denaturation-induced protein damage and delaying protein oxidation and carbonyl production in the pork, with the best effect of 10% FI.

### 3.2. Total Sulfhydryl (SH) Content of MP

The SH group on protein side chains is one of the most active functional groups [13], easily oxidized through dehydrogenation, leading to the formation of disulfide linkages (S-Ss) and subsequently decreasing protein stability [30]. Changes in sulfhydryl groups are imperative for describing how proteins fold, S-Ss form, and proteins are arranged in a particular sample, which in turn are closely linked to oxidation. Therefore, the total SH content is commonly used as a vital indicator for determining the oxidative denaturation of proteins [31]. Figure 2 illustrates that the control group’s total SH content was initially 62.45 nmol/mg, which gradually decreased with each additional F-T cycle, reaching its lowest level of 46.16 nmol/mg after seven F-T cycles. This could be attributed to the sensitivity of sulfhydryl groups in meat protein towards F-T cycles, which results in the formation of intramolecular or intermolecular disulfide linkages [32]. MP is abundant in SH groups, which are easily changed to S-Ss during the oxidative process. According to Wang et al. [30], the formation of disulfide linkages can promote protein aggregation and disrupt protein conformation and spatial structure. Notably, in comparison to the control group, FI groups had a greater total SH content in each F-T cycle. In particular, the total SH content in the 10% FI group only decreased by 8.42% after seven F-T cycles. This may be because FI overlaps with the SH group of the actomyosin molecule, resulting in a decline in oxidation sensitivity to the SH group, thereby effectively delaying the conversion of SH groups to disulfide linkages and stabilizing the protein structure. Additionally, FI contains strong antioxidant active substances that interact with proteins, effectively inhibiting the generation of disulfide linkages and preventing the aggregation of proteins. Furthermore, there was no discernible difference in the total SH content between 10% FI and 0.02% BHA groups throughout the F-T cycles (*p* > 0.05), with comparable effects. This indicates that FI has the potential to replace synthetic antioxidants, and it has a positive effect on delaying protein oxidation and improving the oxidative stability of ground pork in multiple F-T cycles.

### 3.3. Ca^2+^-ATPase and K^+^-ATPase Activities of MP

The Ca^2+^-ATPase and K^+^-ATPase changes in its activities reflect the structural changes that occur after myosin oxidation. Therefore, they are widely used as indicators of the molecular integrity of myosin [33], and the lower the activity, the lower the protein quality. As depicted in Figure 3, the ATPase activities of the MP samples gradually decreased as the F-T cycles increased. Especially, after seven freeze–thaw cycles, the Ca^2+^-ATPase and K^+^-ATPase activities decreased by 58.46% and 61.18% in the control group, respectively. The outcome demonstrated that the protein integrity of the sample was seriously damaged and degraded. The outcome was in line with Xia et al. [22], which demonstrated that the Ca^2+^-ATPase and K^+^-ATPase activities of MP in porcine muscle were decreased during five F-T cycles. This can be explained by the conformational changes and the aggregation of the myosin globular head after multiple F-T cycles [34]. The protein rearrangement due to protein–protein interaction during the freeze–thaw process may also lead to the loss of ATPase activities [35]. In addition, it was possible that the myosin molecule’s proteolysis was connected to the loss of ATPase. What is more, the SH reactivity in the globular head of myosin is intimately correlated with the ATPase activities [36], and its oxidation causes changes in the ATPase activities, thereby explaining the concurrent decline in SH levels and ATPase activities. After seven F-T cycles, the ATPase activity was the highest in the 10% FI group, which was significantly different from that of the control group (*p* < 0.05), and comparable to that of the 0.02% BHA group (*p* > 0.05). This suggested a greater degree of denaturation in myosin-based muscle proteins within the control group, which can be related to the intrinsic structural alterations caused by changes in hydrogen, hydrophobicity, disulfide, and ionic bonds in proteins in multiple F-T cycles. Moreover, the addition of FI and BHA could effectively protect the ATPase activities in MP, so as to maintain the complete structure of protein. The decline in ATPase activities is inevitable during multiple F-T cycles, while the addition of 10% FI can effectively inhibit protein oxidation and provide a certain improvement to the integrity of protein structure, which has a positive significance for the maintenance of MP structure.

### 3.4. Secondary Structure of MP

The Fourier transformation infrared (FTIR) spectra were used to characterize the secondary structure of MP [37]. The amide I band (1600–1700 cm^−1^) is highly sensitive to the secondary structure of the protein, and is used to analyze the aggregation, folding, and unfolding of the protein by measuring the variation in its peak intensity [38,39]. Among them, a higher proportion of α-helices and β-sheets indicated a protein structure that was more stable, whereas a higher proportion of β-turns and random coils indicated a structure that was more relaxed [38]. According to Figure 4, as the F-T cycle increased, the relative α-helix content of each group gradually decreased, especially in the control group, which decreased by 15.22% after seven F-T cycles. In comparison, the random coil content revealed a different trend. The results showed that the loss of the α-helix structure in MP was severe during multiple F-T cycles. Hydrogen bonding between carbonyl and amino groups plays a major role in stabilizing the α-helix structure [40]. It has been reported that the primary causes of the MP conformational changes throughout the F-T cycle are the production and the enlargement of ice crystals [41]. This leads to hydrogen bonding and unfolding of the protein molecular chain, with the result that the MP’s α-helix structure unfolds. And with the repeated F-T cycles, the protein molecules’ continuous expansion, the hydrogen bonds that maintain the MP structure’s stability were damaged, which further lead to a breakdown in ordered structure and an increase in disordered structure. It implies that repetitive F-T cycles induce a shift in MP conformation from an ordered to a disordered flexible framework [42]. What is more, another factor contributing to the destruction of MP secondary structure is the protein oxidation induced by F-T cycles [43]. Compared to the control group, FI is able to effectively stabilize the secondary structure of MP, with 10% being the most effective, with a decrease of 8.27% in the α-helix content and an increase of only 5.43% in the random coils content after seven F-T cycles. This showed that FI might inhibit ice crystal growth, interact with MP to strengthen the intermolecular or intramolecular hydrogen bonds of the protein, and stabilize the structural changes of MP in the F-T cycles. In addition, the previous results showed that 10% FI also significantly inhibited the increase in carbonyl content (Figure 1) and the decrease in total SH content (Figure 2) of MP in ground pork, effectively inhibiting protein oxidation. Thus, the stability improvement in protein secondary structure caused by FI may also be related to its own strong antioxidant activity.

### 3.5. The Protein Solubility of MP

One of the most important indicators for assessing protein functionality is protein solubility, which is closely related to intermolecular forces [44]. It represents the level of protein degradation and is directly connected to the processing and quality of ground meat products after repeated F-T cycles [42]. As shown in Figure 5, the solubility gradually declined with the increase in F-T cycles, and especially reduced significantly after five F-T cycles. MP is a complex protein, that is soluble in neutral salt solutions with an ionic strength greater than or equal to 0.5, hence the name salt-soluble proteins. Salt-soluble protein loss indicates that MP is damaged during the F-T process. A variety of factors may contribute to the drop in MP solubility. In the F-T cycles, the structural stability of MP decreases, and some MP precipitates can form ice crystals after combining with water, which will further lead to the formation of hydrogen bonds or hydrophobicity between actin molecules. As a result, the insoluble agglomerates of large macromolecules are formed, which continuously reduces MP solubility [45]. The alkali-soluble proteins produced by MP denaturation are not soluble at high ionic strengths, and this contributes to the reduced solubility of actinomyosin in repeated F-T cycles. And the repeated F-T cycles could turn liquid water into solid ice, resulting in an increase in the concentration of intracellular ion, which causes proteins to oxidize and denature and further results in a reduction in protein solubility [46]. In addition, the oxidation of the SH group also forms S-Ss, which cross-link and aggregate proteins and reduce their solubility [47]. Du et al. [48] proposed that the exposure of hydrophobic groups and SH groups promoted intermolecular cross-linking of protein aggregates as the protein structure unfolded, which was closely associated with a decrease in MP solubility. This is consistent with the findings of the total SH content variation (Figure 2). After seven cycles, the solubility of MP in all six groups decreased by 29.69%, 28.14%, 22.86%, 22.00%, 23.77%, and 25.03%. Protein denaturation that occurs during freezing may lower the meat’s ability to hold water, which could result in the development of thaw loss. The addition of FI significantly delayed the decrease in MP solubility in ground pork samples during F-T cycles (*p* < 0.05), and the 10% FI group had the best effect. This may be because the addition of FI delayed solute migration during freezing, reduced ionic strength in unfrozen liquids, as well as inhibited cross-linking and aggregation of proteins by disrupting their hydrophobicity. Next, the addition of FI delayed the oxidation of proteins to some extent, increased the total SH content, and stabilized the structure of proteins.

### 3.6. Emulsifying Properties of MP

The oil and water retention of meat products is greatly affected by the emulsification properties of proteins, which has an impact on the yield and quality of the product. Protein emulsification properties are commonly evaluated using the EAI and ESI [49]. EAI represents the capacity of the protein to stay at the oil–water interface to prevent flocculation and aggregation following emulsion formation, which reflects the complex interactions between protein–protein and protein–lipid [50,51]. As illustrated in Figure 6A, the EAI of the six groups gradually decreased as the F-T cycles increased. In particular, a considerable decline was seen after three F-T cycles. The cycles of freezing and thawing promote protein denaturation, increasing the protein particle size, which is a key factor contributing to the reduction in EAI. Moreover, protein aggregation may produce larger proteins, increasing the space between the water and oil in the emulsion, thereby reducing the potential of protein molecules to adsorb to the oil–water interface [52]. The control group exhibited the most significant decline in EAI after seven F-T cycles, with a decrease of 10.04%, which may be due to the fact that the proteins produced by oxidation are too huge to remain flexible when adsorbed to the oil droplet’s surface. In the study conducted by Marwan et al. [53], the solubility and emulsifying ability of sarcoplasmic proteins were significantly decreased due to the oxidation products generated during protein storage. However, FI effectively prevented the decrease in EAI, in which the addition of 10% FI had the best effect with only a 7.07% decrease. The above results demonstrate that FI maintains the protein structure during the F-T cycle, which would promote protein adsorption on the interfacial layer and positively affect the emulsification properties of the protein.

The ESI quantifies the protein’s ability to stabilize the emulsion dispersion system, with higher values indicating a reduced likelihood of phase separation between water and oil in the emulsion [54]. It can be seen from Figure 6 that ESI and EAI both exhibit the same tendency. The reduced ESI may be attributed to protein aggregation and decreased flexible peptides migrating to the oil-water interface due to oxidation during F-T cycles. Additionally, the oil droplets’ fluidity is decreased due to the protein aggregation that is caused by oxidation being adsorbed on their surface. The addition of FI also inhibited the decrease in ESI, and the effect was superior to those of the NWP and BHA groups. In particular, after seven F-T cycles, the addition of 10% FI considerably prevented the decrease in ESI, which increased by 9.1% compared to the control group. It is evident that FI contributes to maintaining the MP emulsion system stable during F-T cycles. A research study has indicated that MP oxidative denaturation was the primary element determining emulsification capacity, which may also be impacted by various storage and processing circumstances [55]. What is more, the protein–oil and protein–water interactions are also necessary for the emulsifying characteristics of proteins. Protein needs to interact strongly with water in order to be a good emulsifier, which suggests that it has a high solubility. The previous findings indicated that the 10% FI sample had higher solubility (Figure 5), which effectively enhanced protein–oil and protein–water interactions. Combined with the above results, it was found that 10% FI might lessen the degree of protein oxidation by decreasing the exposure of hydrophobic groups and SH on the surface of proteins, and play a role in protecting the emulsification properties. In addition, the addition of 10% FI also reduces the mechanical harm of ice crystals in the muscle tissue during F-T cycles, while also playing a pivotal role in mixing and cross-linking with MP, which led to a reduction in the extent of MP aggregation and ultimately improved the emulsification properties of MP.

## 4. Conclusions

In the current study, repeated F-T cycles led to oxidative aggregation and denaturation of proteins, thereby causing a decrease in the MP’s conformational stability and functional properties. Significant differences in MP oxidation, structural, and functional properties were observed in minced pork with different amounts of FI additions. As we expected, FI inhibited the formation of carbonyl and disulfide linkages to a certain extent and effectively protected the ATPase activity in MP compared to the control group and other treatment groups. Thus, the secondary structure of MP was stabilized, and the solubility and emulsification properties of MP were improved, and the optimal additive amount was 10%. The results showed that 10% FI played an important role in inhibiting the rate of protein oxidation and improving the structural stability and functional properties of MP during the F-T cycle process. The current study contributed positively to exploring the role of FI in inhibiting oxidative chain reactions in ground pork through its antioxidant action, which slowed down protein denaturation caused by oxidation and reduced the rate at which repeatedly freeze–thawed ground pork undergoes oxidation. These results will also provide a theoretical support for the application of natural antioxidant peptides in the oxidation prevention and the quality assurance of ground pork products.

## Figures and Tables

**Figure 1 foods-12-03135-f001:**
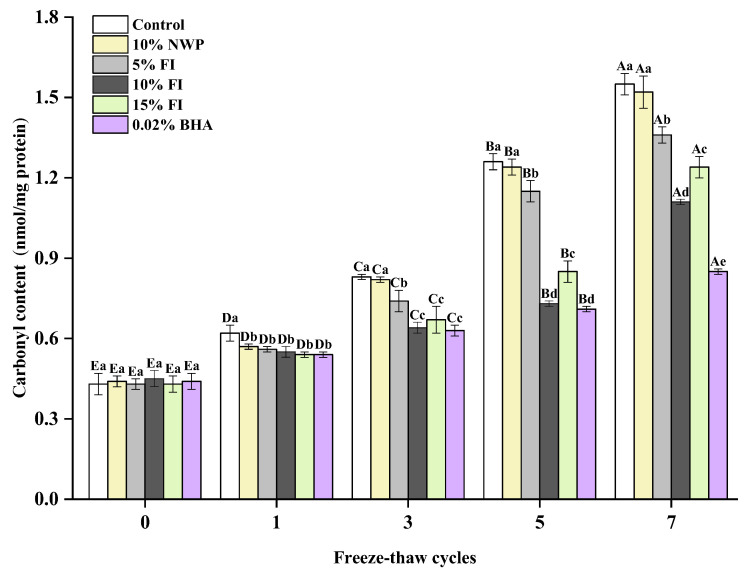
Changes in the myofibrillar protein of carbonyl contents with different fraction I (FI) contents in ground pork during repeated freeze–thaw (F-T) cycles. Significant differences between 0, 1, 3, 5, and 7 F-T cycles are indicated with different capital letters (A–E). Significant differences among different samples are indicated by different lowercase letters (a–e). FI: 4-h whey protein hydrolysate with the best antioxidant activity (<1 kDa); Control: no additives in the sample; NWP: native whey protein isolate; BHA: butylated hydroxyanisole. The same as below.

**Figure 2 foods-12-03135-f002:**
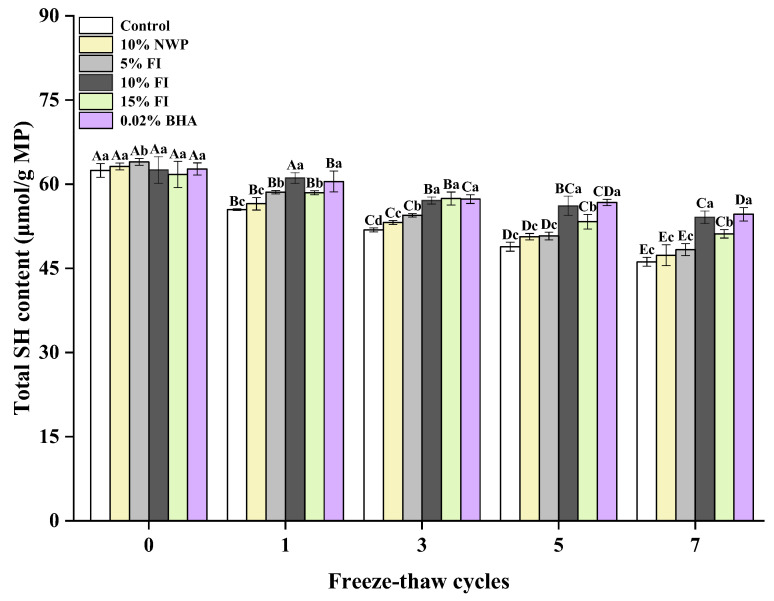
Changes in the myofibrillar protein of total sulfhydryl contents with different fraction I (FI) contents in ground pork during repeated freeze–thaw (F-T) cycles. Significant differences between 0, 1, 3, 5, and 7 F-T cycles are indicated with different capital letters (A–E). Significant differences among different samples are indicated by different lowercase letters (a–d). FI: 4-h whey protein hydrolysate with the best antioxidant activity (<1 kDa).

**Figure 3 foods-12-03135-f003:**
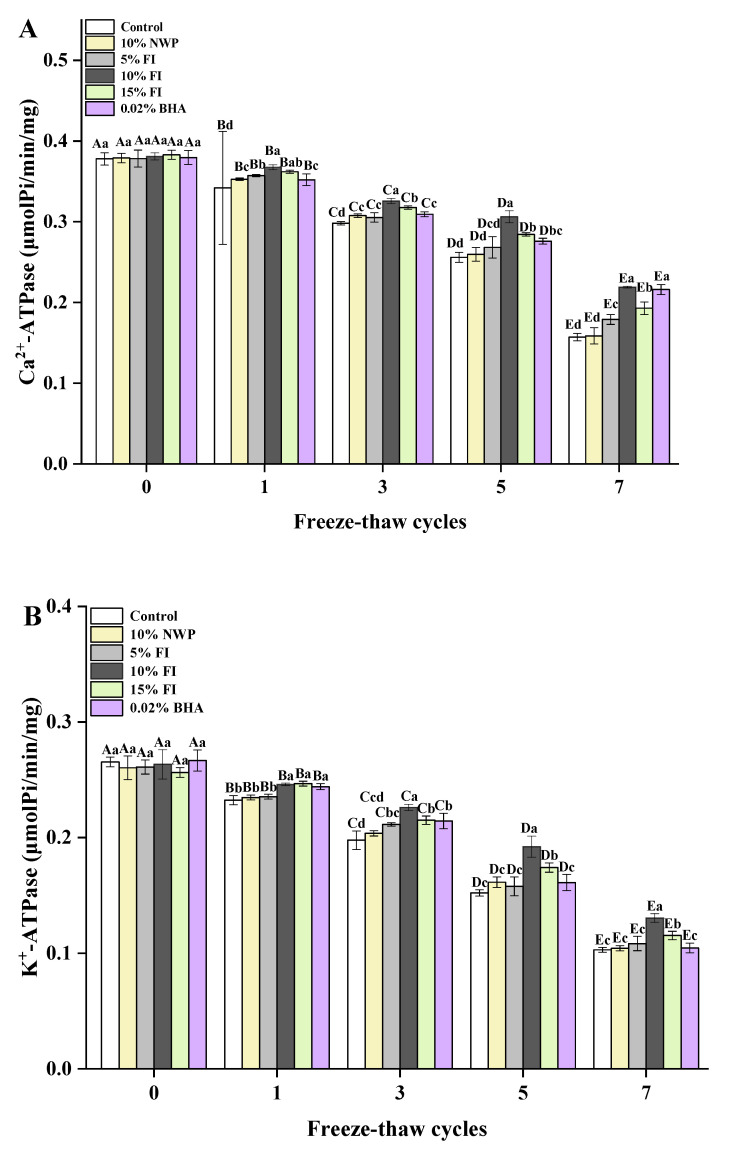
Changes in the myofibrillar protein of Ca^2+^-ATPase (**A**) and K^+^-ATPase (**B**) with different fraction I (FI) contents in ground pork during repeated freeze–thaw (F-T) cycles. Significant differences between 0, 1, 3, 5, and 7 F-T cycles are indicated with different capital letters (A–E). Significant differences among different samples are indicated by different lowercase letters (a–d). FI: 4-h whey protein hydrolysate with the best antioxidant activity (<1 kDa).

**Figure 4 foods-12-03135-f004:**
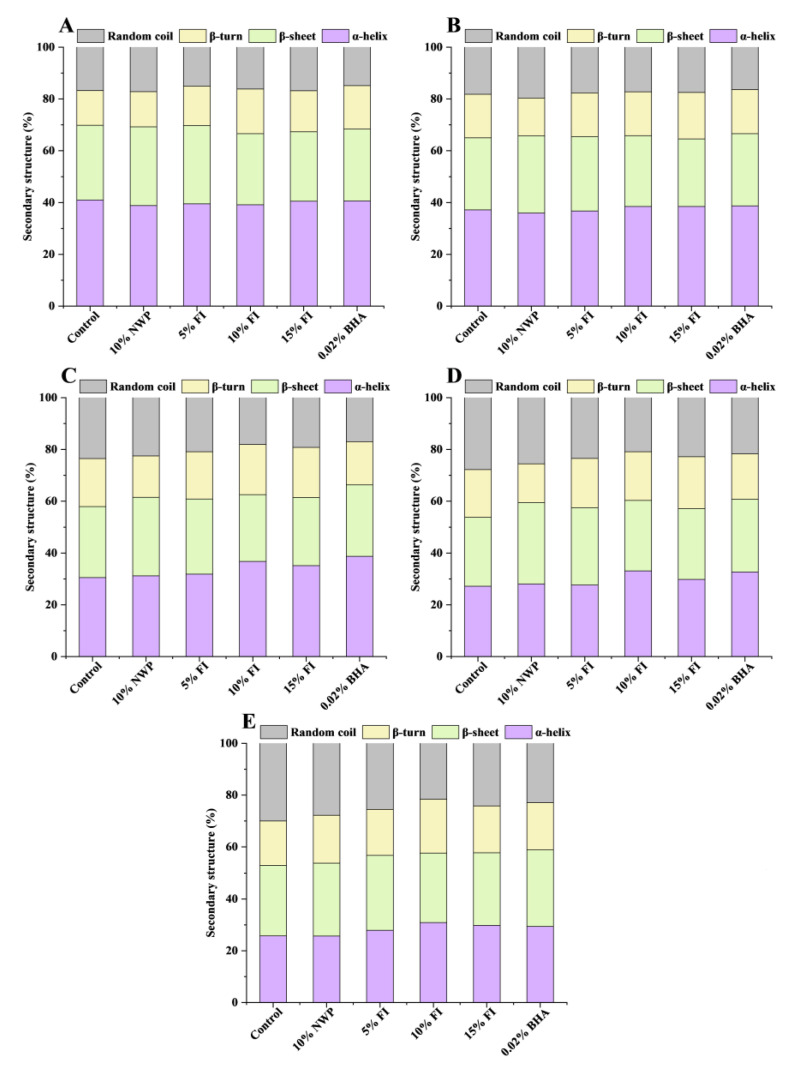
Changes in the myofibrillar protein of secondary structure with different fraction I (FI) contents in ground pork during repeated freeze–thaw (F-T) cycles. (**A**–**E**) denotes 0, 1, 3, 5, and 7 F-T cycles. FI: 4-h whey protein hydrolysate with the best antioxidant activity (<1 kDa).

**Figure 5 foods-12-03135-f005:**
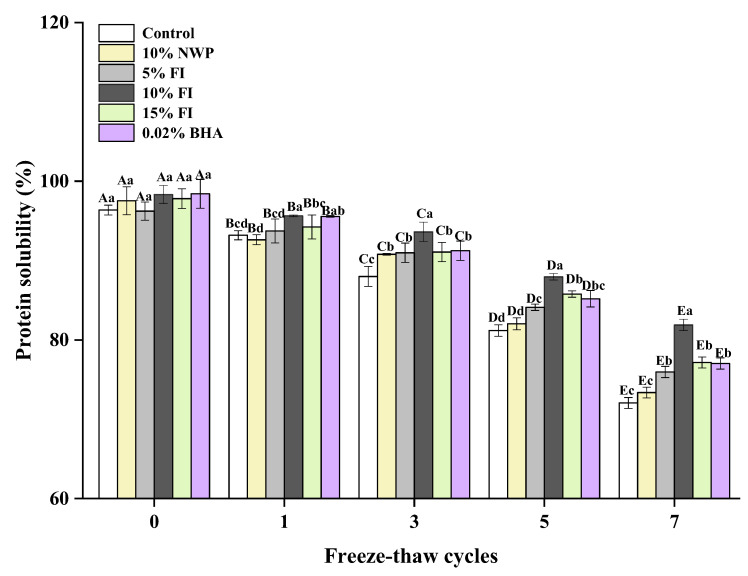
Changes in the myofibrillar protein of solubility with different fraction I (FI) contents in ground pork during repeated freeze–thaw (F-T) cycles. Significant differences between 0, 1, 3, 5, and 7 F-T cycles are indicated with different capital letters (A–E). Significant differences among different samples are indicated by different lowercase letters (a–d). FI: 4-h whey protein hydrolysate with the best antioxidant activity (<1 kDa).

**Figure 6 foods-12-03135-f006:**
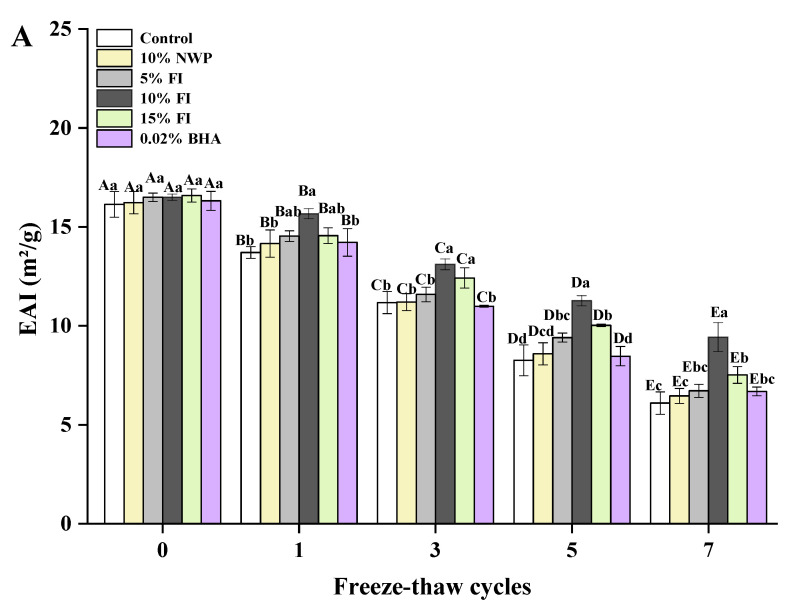
Changes in the myofibrillar protein of emulsification activity index (**A**) and emulsion stability index (**B**) with different fraction I (FI) contents in ground pork during repeated freeze–thaw (F-T) cycles. Significant differences between 0, 1, 3, 5, and 7 F-T cycles are indicated with different capital letters (A–E). Significant differences among different samples are indicated by different lowercase letters (a–d). FI: 4-h whey protein hydrolysate with the best antioxidant activity (<1 kDa).

## Data Availability

Data are contained within the article.

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
