# Peer review of "Whey Protein Hydrolysate Improved the Structure and Function of Myofibrillar Protein in Ground Pork during Repeated Freeze–Thaw Cycles"

_foods, 2023, doi:10.3390/foods12163135_

Round 1

Reviewer 1 Report

Abstract

Clearly state the objective of the study in the first sentence. For example, "The objective of this study was to investigate the effect of whey protein hydrolysate-frac- tion I (FI, the molecular weight < 1 kDa) on the frame and function of myofibrillar protein (MP) in ground pork during repeated freeze-thaw (F-T) cycles. The current abstract contains some lengthy sentences that could be shortened for clarity and readability. Break down complex ideas into simpler sentences to improve understanding.

Introduction

Begin the introduction by providing a brief overview of the importance of freezing preservation in meat products and the occurrence of multiple freeze-thaw (F-T) cycles during the storage, production, transit, and sale of meat products.

Emphasize the adverse effects of F-T cycles on the physical and chemical properties of meat products, such as poor color, texture deterioration, flavor loss, and water loss, leading to significant economic losses.

Explain that the functional properties of muscle protein, such as solubility, emulsification, and water binding capacity, play a crucial role in enhancing the texture and moisture retention of meat products. Highlight the need to stabilize the structure and improve the functional properties of MP during F-T cycles to produce high-quality ground pork products with an extended shelf life.

Methods and Materials

2.5 what is Carbonyl content? Does this study really need this parameter? Justify please.

2.6 Total sulfhydryl (SH) content have been measured, and no hydrophobicity measurement?

No emulsion rheology and particle size has been determined, while these are the basic study to evaluate any emulsion properties.

Results

All of the graphic figures are not clear, hardly to understand the difference, change the color and also visualization of the figure, these are not up to the mark.

Secondary structure must be presented by shift of curves in analysis, as well as with table data to see the actual secondary structure of protein, provide this please.

All more of discussion in each part of results

Conclusion

Please check the conclusion, minimize the sentences and add few practical applications of this study.

Reviewer 2 Report

Please revise the English language and grammar to improve the readability 

Author Response

请参阅附件。

Reviewer 3 Report

The manuscript “Effect of whey protein hydrolysate with the molecular weight less than 1 kDa on the structure and function of myofibrillar protein in ground pork during repeated freeze-thaw cycles” presents a set of relevant results on the protein changes occurring during several freezing-thawing cycles and the protective effect of whey protein hydrolyzates.

Abstract

Page 1, lines 12-14 – Please revise this sentence for clarification.

Page 1, line 20 – I think it is “upgraded”.

Page 1, line 21 – I suggest “preserving” instead of “enhancing”.

Introduction

Page 1, line 30 – It is doubtful that freezing preservation could improve quality.

Page 1, line 42 – I suggest “muscle proteins”.

Page 1, line 43 – Please check the sentence: “based on muscle… characteristics” for clarification.

Page 2, lines 66 and 67 – The sentence “Among them, … (NWP)” is also not clear. Please check.

Page 2, line 83 – I suppose it is “popularization” instead of “popularize”.

Materials and Methods

Page 2, line 90 – I suggest replacing “contained” with “included”.

Page 3, line 97 – I also suggest replacing “formulated” with another more adequate word.

Page 3, line 99 – I think it is “kept” instead of “preserved”.

Page 3, line 103 – The preparation of FI fraction (<1 kDa) is not mentioned. Please check.

Page 3, lines 104 and 105 – The levels of hydrolysate used are not reported. Please check.

Page 3, line 107 – I suggest “pork longissimus dorsi”.

Page 3, line 130 – Please clarify what is meant by “equivalent”.

Page 3, line 132 – Please indicate the speed of centrifugation.

Page 3, line 133 – I suggest “pellets” instead of “precipitations”.

Page 3, lines 135 and 136 – Please indicate the speed of centrifugation.

Page 4, line 146 – Please indicate the volume of the MP samples.

Page 4, line 165 – Please clarify the sentence “Before and after… centrifuged”.

Page 4, line 175 – I suggest replacing “obtained” with other word such as “collected”, for instance.

Page 4, lines 175 and 176 – Please consider this alternative: “…and dispersed in 5 mL of the solution with 0.1 % sodium dodecyl sulphate (SDS) solution”. I also suggest “The 0.1 % SDS solution was used as a...”

Page 5, lines 200-204 – Please consider the following alternative sentence: “This can be attributed to the disruption of the microstructure and integrity of muscle cells by the water recrystallization occurring during repeated F-T cycles. These physical changes of the muscle cells promote the release of prooxidants and thus exacerbating the protein oxidation and accelerating the formation of carbonyl groups [27].”

Page 7, line 270 – Please replace “easing” with another more adequate word.

Page 9, line 313 – I suggest replacing “vital metrics” with another words and I think it is “functionality” instead of “function”.

Page 10, lines 315 and 316 – Please consider this alternative: “…the mechanical harm of ice crystals in the muscle tissue…”

Page 10, line 371 – I suggest replacing “a large” with “higher”.

Page 11, line 384 – I suggest replacing “deepening” with another word.

References

Please revise the references 16, 29, 34, 36, 39, 44, 25, and 52 because they are not correct.

The English language needs some moderate edition.

Round 2

Reviewer 2 Report

Thank you for your thorough revision! Here are some last comments:

- Please delete the abbreviation in the title and use present tense 

- In the Conclusion the information is still missing, how this paper extends current knowledge 
